# Risk Assessment and Area-Wide Crop Rotation to Keep Western Corn Rootworm below Damage Thresholds and Avoid Insecticide Use in European Maize Production

**DOI:** 10.3390/insects13050415

**Published:** 2022-04-27

**Authors:** Lorenzo Furlan, Francesca Chiarini, Barbara Contiero, Isadora Benvegnù, Finbarr G. Horgan, Tomislav Kos, Darija Lemić, Renata Bažok

**Affiliations:** 1Veneto Agricoltura, Agricultural Research Department, Viale dell’Università, 14, 35020 Legnaro, PD, Italy; francesca.chiarini@venetoagricoltura.org; 2Department of Animal Medicine, Production and Health, University of Padua, Agripolis, Viale dell’Università, 16, 35020 Legnaro, PD, Italy; barbara.contiero@unipd.it; 3Via G. Mameli 13, 45011 Adria, RO, Italy; isadora.benvegnu@gmail.com; 4EcoLaVerna Integral Restoration Ecology, Bridestown, Kildinan, T56 P499 County Cork, Ireland; f.horgan@ecolaverna.org; 5Faculty of Agrarian and Forestry Sciences, School of Agronomy, Catholic University of Maule, Casilla 7-D, Curicó 3349001, Chile; 6Centre for Pesticide Suicide Prevention, University/BHF Centre for Cardiovascular Science, University of Edinburgh, Edinburgh EH16 4TJ, UK; 7Agronomy and Aquaculture, Department of Ecology, University of Zadar, Prince Višeslav Square 9, 23000 Zadar, Croatia; tkos@unizd.hr; 8Faculty of Agriculture, University of Zagreb, Svetošimunska 25, 10000 Zagreb, Croatia; dlemic@agr.hr

**Keywords:** IPM, risk factors, structural crop rotation, flexible crop rotation, damage threshold, area-wide strategies

## Abstract

**Simple Summary:**

*Diabrotica virgifera virgifera* LeConte, the Western corn rootworm (WCR), is a maize-specific pest that has been a serious threat in Europe since the mid-1990s. To properly implement integrated pest management, it is necessary to identify the key factors associated with risks of crop damage from WCR and to evaluate the effectiveness of area-wide strategies based on agronomic measures, such as crop rotation, in reducing those risks. In Italy and Croatia, a survey of agronomic and cultural factors in fields damaged by WCR allowed us to determine that the beetle population size accounts for most of the risk of maize damage from WCR. Crop rotation (without insecticide use), both structural and flexible, was the most effective strategy for keeping WCR populations below the damage threshold. This indicates that WCR management can be carried out in accordance with European Union regulations to limit or avoid insecticide treatments and reduce environmental impacts.

**Abstract:**

The Western corn rootworm (WCR), *Diabrotica virgifera virgifera* LeConte, has been a serious quarantine pest to maize in Europe since the mid-1990s. The integrated pest management of WCR requires an accurate knowledge of the factors that contribute most to risks of crop damage, as well as knowledge of effective area-wide strategies based on agronomic measures, such as crop rotation. In Italy and Croatia, agronomic and cultural factors in fields damaged by WCR were evaluated through a long-term survey. Based on the survey results, high-WCR densities contribute most to risks of damage to maize. Extensive field research in north-eastern Italy compared large areas of continuous maize production with areas under different crop rotation systems (i.e., a structural one with one-time maize planting in a three-year rotation and a flexible one with continuous maize planting interrupted when beetle populations exceed the threshold). The objective was to evaluate the effectiveness of different rotation regimes as possible best practices for WCR management. Captures of beetles in yellow sticky traps, root damage, larval densities, and damage to maize plants (e.g., lodging) were assessed at the center of each area. The results demonstrated the both structural and flexible crop rotation systems were effective strategies for maintaining WCR below damage threshold densities without the need for insecticides.

## 1. Introduction

The most serious maize pest in North America [1], the Western corn rootworm (WCR) (*Diabrotica virgifera virgifera* LeConte; Coleoptera: Chrysomelidae), is now established in most European maize growing regions [2]. In the last twenty-five years, a significant amount of new knowledge about WCR in Europe has been gained by the scientific community. This knowledge was summarized by Bazok et al. [2] and included information on the monitoring and spread of WCR, on its ecology and damage, and on control methods and management tools, including tools for monitoring and biological control, as well as predictions of further spread and damage. During the last 10 years, WCR has been considered a naturalized pest in many European Union (EU) countries, as it has become a regular part of the entomofauna [2]. There are a number of differences between agricultural systems in the EU and agricultural systems in the USA, where this pest causes the most damage. The approach to WCR control in the EU has, therefore, been adapted to agricultural systems and legislation.

Integrated pest management (IPM) as a cornerstone of sustainable agriculture is strongly advocated by EU policy [3]. IPM aims to improve farmers’ practices to achieve higher profits while reducing pesticide use. Successful implementation of IPM will require novel control methods and new strategies that reduce the current reliance on insecticides. The first step in IPM is prevention, i.e., the implementation of a series of agronomic measures such as crop rotation, and where appropriate the use of resistant or tolerant varieties, which create conditions that reduce the risk of pest outbreaks and the need for curative plant protection measures. Crop rotation is the first preventive measure listed in Annex III of Directive 128/2009/EC [3]. Since the most effective strategy against WCR is crop rotation [2,3,4,5,6], its implementation is mandatory under this legislation. Crop rotation can be implemented as a long-term structural measure (e.g., using winter wheat, maize, soybeans) or with flexible modalities. According to the flexible method, to maintain forage production on livestock farms at optimal levels in terms of yield and quality, maize can be grown continuously for two or more years, with other crops planted on a flexible schedule if adult WCR densities increase significantly, and especially if the damage threshold is exceeded. Captures of ≥6 beetles/Pherocon AM trap/day over a 6-week period (or even less) as previously recommended in USA has proven to be an effective WCR threshold density in Europe [7,8].

As described many times in the literature, the first records of WCR in Europe were reported from Serbia in the early 1990s [9]; however, WCR is believed to have been introduced into Europe in the early 1980s [10]. After the establishment period, WCR began to cause economic damage to maize production in Croatia and neighboring countries in the 1990s [10,11]. Based on population genetic studies of WCR by Miller et al. [12] and Ciosi et al. [13,14], the population source for the introduction to Serbia was probably Pennsylvania (USA). Based on analysis by Lemic et al. [15] of the genetic variability of WCR populations soon after the introduction in the 1990s and again in the period between 2009 and 2011, control measures (e.g., crop rotation) initiated soon after the introduction did not result in genetic differentiation toward management-adapted populations, as had previously occurred in USA [16]. The results from a range of genetic studies [10,11,12,13,14,15,16] suggest that there have been several introductions of WCR into the Veneto region of Italy, where the populations intermingled: in the Veneto region (Venice airport outbreak area), WCR populations had been successfully eradicated [17], but later reinvaded from the east and west, making Veneto an area where WCR populations from different parts of the USA have mixed [18,19]. These included the first reported rotation-resistant strain. Therefore, understanding the population dynamics of WCR in the Veneto region will be particularly useful as a model for other regions with evolving WCR populations.

The primary objective of the present study is to provide information useful for area-wide management of WCR so that maize can be grown without negatively impacting farmers, in compliance with IPM, and without insecticide use. Our specific objectives were to (i) analyze the risk factors for WCR damage to maize and (ii) evaluate the success of structural and flexible crop rotation as a significant component of best management practices for WCR in Europe.

## 2. Materials and Methods

Two main extensive and long-term surveys were carried out. These were:A WCR risk assessment survey that included damaged fields in both Italian and Croatian maize cultivation areas (Figure 1) and that considered a large number of potential risk factors;A WCR area-wide management survey in northern Italy where some maize cultivation areas under different rotation approaches (structural or flexible rotation) and pest control practices were monitored to assess the effects of these practices on WCR populations and on the consequent risks of damage.

### 2.1. WCR Risk Factors Study in Italy and Croatia

A comprehensive survey of maize fields damaged by WCR was conducted in north-eastern Italy (area covered: 46°06′ N, 12°00′ E and 45°21′ N, 11°38′ E) from 2010 to 2017 and in Croatia (area covered: 46°23′ N, 19°10′ E and 45°15′ N,16°21′ E) from 2003 to 2014 (Figure 1). This produced a large set of infestation and damage data (Table 1). Some additional fields in the vicinity of the damaged fields were additionally surveyed. In Italy, the 1126 ha of maize surveyed from 2010 to 2017 included a total of 602 fields. In Croatia, 143 fields were surveyed. Depending on the year, the number of fields surveyed varied from one to thirty, and the total area surveyed in Croatia was 139.26 ha. The total cultivated area of the farms involved in the study was 2882 ha. The main variables recorded in the extended survey are shown in Table 1.

#### 2.1.1. Soil Properties

Soil texture data for Italy were obtained from the Veneto Region Environmental Protection Agency (ARPAV) database [20]. The soils of each surveyed field were classified based on soil texture, according to the soil characteristics of corresponding soil map units (SMU) (see ARPAV database [20] for more details). Soil texture was determined using the United States Department of Agriculture (USDA) triangular method [21] based on analysis with a sediment pipette. The soils were loam, clay loam (F), loamy sand, sandy clay loam; silt loam, silty clay loam (FL), clay loam (FA), silty clay loam, silty clay (FLA), and sandy loam (FS) (Table 1). For statistical analyses FA and FLA were grouped together. Soil texture data for Croatia were taken from the database of the interactive soil map of Croatia created for Google Earth [22].

#### 2.1.2. Agronomic Practices

Land management practices were similar at all study sites and included: fertilizer applications at 240–300 kg N/ha; 70,000 to 80,000 seeds/ha; and 75 cm inter-row width in Italy and 70 cm in Croatia. Seeding depth varied from 2.5 to 9 cm (dry seedbed) at the study sites. All sampled fields in Croatia and the majority of sampled fields in Italy were conventionally tilled (i.e., plowing, cultivating, harrowing, and hoeing), while approximately one out of every four fields under structural rotation in Italy were minimally tilled (i.e., one cultivator pass, harrowing, and hoeing). Weed incidence was low and most farmers had applied pre-emergence and post-emergence herbicides.

##### Rotation Regimes

For the extensive survey, rotation regimes were categorized into three types as follows. Rotation type A: Continuous maize cultivation (for at least four years) even as a second crop after early harvested main crops such as ryegrass; maize accounted for more than 80% of the crops in the rotation. Rotation type B: Maize was the predominant crop (≥40% of crops in rotation, very often 70–80%), often as a permanent crop, but sometimes alternated with different crops in a flexible sequence according to WCR densities, with soybean, winter cereals, and sorghum being the most commonly rotated crops. Rotation type C: Maize was grown once or twice in the period considered and accounted for 30–40% of the crops in the rotation; many other crops were grown as in B, including double crops (e.g., soybean or sorghum after barley or canola) or meadows (e.g., alfalfa, *Festuca* spp., etc.), while continuous maize cultivation was very rare.

##### Sowing Date

All fields were divided according to sowing date into four categories as early (March), ordinary (April), late (May), and very late.

##### Hybrids

The following commercial maize hybrids were the main ones grown (≥95% of the study plots): in Italy, DKC5276, 5830, 6815, 6718, P1028, P1547, DKC5830 (2015–2019), P1028, P1114, PR32B10, A14, Y43, Kerbanis, LG 33.30, 34.09, and 34.10; in Croatia, BC hybrids (BC Institute) BC 566, 578, 678, 354, 462B, 462, Jumbo 48, 408B, 4982, Pioneer hybrids Colomba, Natalia, Florencia, PR36V52, PR36R10, 33A46-1007, Dekalb hybrids DK571, Occitan, LG 33.30, and 34.09. To our knowledge, none of these hybrids have any varietal resistance or tolerance to WCR.

#### 2.1.3. Insecticide Treatments

Soil insecticide and seed coating treatments are listed in Table 2. The main insecticides used as foliar treatments against the beetles were: lambda-cychalothrin 9.48% (200 mL/ha); alfacypermethrin 15% (0.4 kg/ha); chlorantraniliprole 9.26% + lambda-cychalothrin 4.63% (300 mL/ha).

#### 2.1.4. Damage Assessment

In Italy, at the end of the development period of the WCR larvae, in each monitored field, after an initial general assessment of the homogeneity of the field, two or four (in case of higher variability of crop damage) subplots of four 20 m maize rows were randomly selected. In cases where the fields had obviously different conditions due to cropping density or plant development, the fields were subdivided into relatively homogenous sections; each section was then assessed by randomly selecting at least two subplots and considering the field sections as separate records in the database. In Croatia, four rows of 100 plants were randomly selected in different parts of the fields. In order to reduce the influence of possible ‘border effects’ [23], only fields that presented visible plant damage exceeding 5% of the plants and distributed throughout the cultivated field were considered for the analysis.

The following parameters were recorded for each subplot (in the case of Italy) or row (in the case of Croatia): (1) the number of normal plants (no symptoms); (2) the number of plants with typical ‘gooseneck’ symptoms (plants leaning to the ground but not touching it and curving upright); (3) the number of lodged plants (plants touching the ground and not curving to an upright position). Total damage from WCR was calculated as the sum of plants with gooseneck symptoms together with lodged plants. To calculate the proportions damaged, the average number of plants with gooseneck symptoms + lodged plants in each subplot or transect was divided by the average number of total plants (plants with gooseneck symptoms + lodged plants + normal plants). The calculated percentage was entered into the database along with the field characteristics. To ensure that goosenecked and lodged plants were each primarily caused by larval feeding, at least 20 root systems per subplot were excavated and examined for WCR and root damage. For root damage assessments before 2010, the Hills and Peters 1–6 root damage rating scale [24] was used, and after that year the Node Injury Scale 0–3 [25] was used.

#### 2.1.5. WCR Population Level

WCR beetle population assessments were performed as per the description in Section 2.2. In 60% of surveyed fields that had signs of damage, we deployed Pherocon^®^ AM (PhAM) traps (Trece Inc., Adair, OK, USA); in the remaining fields, the mean number of adults from at least three of the surrounding (within 500 m) monitored maize fields was considered (24% of the total number of monitored fields). If properly managed traps were not available, the population size was considered not assessed (NA). This category included 16% of the fields.

### 2.2. Crop Rotation Study in Italy

Large areas cultivated mainly using maize rotated with other crops were compared against continuously cultivated maize in Veneto, Italy (Figure 1), to assess the effects of two rotation types on beetle and larval densities, as well as on root and plant damage by WCR (see Section 2.2.1, Section 2.2.2 and Section 2.2.3). We assessed the two main crop rotation strategies, structural rotation and flexible rotation, which are used to manage WCR populations in separate comparative studies. The experiment with flexible rotation was initially established as a randomized block design with three replications. However, because of the large areas involved in the study and the differences between the replicated sites in terms of beetle pressures, climatic conditions, and actual rates of rotated maize fields, comparisons are presented separately for each pair of comparative sites (i.e., IR1 and CH1, IR2 and CH2, IR3 and CH3). Fields (1 to 2 ha in size) where maize had been grown previously (continuous maize fields) or with maize fields in rotation were selected in the middle of croplands (about 100 ha in size) (Figure 1). The maize fields were extensively monitored from 2010 to 2020 (area covered: 45°64′ N, 12°96′ E and 45°05′ N, 11°88′ E). Field conditions were generally homogeneous with respect to their main agronomic characteristics.

The field sites are detailed in Section 2.2.1 and Section 2.2.2. The rotation types are implemented as outlined below.

Structural crop rotation: A regular rotation regime where maize is sown once out of two or preferably more years (e.g., wheat–maize–soybean or wheat–soybean–maize; maize can represent 33% with three crops in rotation, 25% with four crops, and so on);

Flexible crop rotation: In areas where there is an appreciable incidence of continuous maize cultivation, maize fields are monitored using PhAM traps; when trap captures exceed the damage threshold (6 beetles/trap/day over several weeks), maize cultivation would be interrupted in the next year. With this strategy, it is expected that continuous maize cultivation will be interrupted after 2–3 years.

#### 2.2.1. Structural Crop Rotation

Between 2010 and 2020, observations of WCR densities and crop damage were made at the three sites with structural rotation regimes. The first site was located at the experimental farm in Vallevecchia (Venice Province), an isolated area on the north-eastern Adriatic coast (45.05 N, 11.88 E, 2 m a.s.l.). The soil in Vallevecchia is Gleyic Fluvisol or Endogleyic Fluvic Cambisol [26], with textures ranging from silty-loam to sandy-loam. It has the following average characteristics: sand 34.2%, silt 42.6%, clay 23.2%; pH = 8.3; active carbonate = 3.0%; organic carbon = 1.0%. The climate is sub-humid, with annual rainfall of about 700 mm. Eighty percent of the soils are well drained, with the being remainder poorly drained. The Vallevecchia farm had a cultivated area of about 380 ha and a total area of about 600 ha.

The second site was located at the experimental farm in Diana (Treviso Province, 45°35′ N 12°18′ E, 6 m a.s.l.), which has a cultivated area of 70 ha. The third site was located at the center of 190 ha of cultivated land in Sasse Rami (Rovigo Province 45°30′ N 11°53′ E, 2 m a.s.l.) (Figure 1). Soils at both these sites are characterized by Endogleyan Cambisols [26], mainly silty-loam soils, and are more homogeneous in texture than Vallevecchia. The climate in Diana and Sasse Rami is also sub-humid, with annual rainfall rates of 846 mm and 673 mm, respectively. Temperatures in Vallevecchia, Diana, and Sasse Rami vary from January (minimum average: 0.1 °C, 0.9 °C, and 0.2 °C, respectively) to July (maximum average: 29.6 °C, 29.3 °C, and 30.6 °C, respectively).

A continuous structural crop rotation regime (winter wheat–maize–soybean) was applied at the sites, with some minor variations for experimental needs or because of constraints due to climatic conditions. Maize-after-maize accounted for about 7 to 10% of the cropped area in the first years when some fields were continuously planted with maize (mainly aimed at obtaining as much forage as possible from maize as the main crop). The two cropping regimes were compared for WCR populations as described above.

After a significant increase in the WCR population was detected in 2016 in Vallevecchia, the proportions of continuous maize fields were reduced to 3–4%. At all farms, about half of the fields were conventionally cultivated, i.e., plowed, cultivated, harrowed, and hoed, while the remaining fields were cultivated using conservation farming methods (minimum tillage or no tillage).

The following common agronomic practices were applied in all of the studied fields: fertilizer applications of 240–300 kg N/ha; 70,000 to 80,000 seeds/ha; 75 cm inter-row width; pre-emergence and post-emergence herbicide treatments.

At each site, at least two fields per year were selected for beetle monitoring using traps and root sampling (2016 to 2018 only), while 10 fields were always assessed for symptoms of crop damage using the methods described below.

#### 2.2.2. Flexible Crop Rotation

A two-year experiment was conducted in 2016–2017 in the province of Treviso in an area with intensive maize cultivation in the Veneto region (45°64′ N, 12°96′ E). The experiment compared crop rotation as a crucial IPM tool with conventional management under long-term continuous maize cultivation and using insecticides to control WCR.

Two homogeneous areas (scenarios) of about 100 ha each were selected in three zones with heavy maize cultivation, consisting of one or more farms; the two homogeneous areas in each zone were characterized by: Intensive rotation (agronomic control with intensive rotation) (IR): Maize generally rotated every two years (maize area 66% per year), with no prolonged maize cultivation; monitoring of pests with traps; no use or reduced use of chemical treatments;Chemical approach (chemical control) (CA): Presence of continuous maize plots over many years with adult pest treatments or seed treatments at seeding with insecticides on most or all of the cultivated area.

Based on preliminary surveys conducted during the 2016 and 2017 maize seasons, three pairs (replicates) of regimes (CA and IR) were concretely identified, each at least one kilometer apart. The first pair (1) comprised two main municipalities, Montebelluna (CA1) and Trevignano (IR1); the second pair (2) was located inside the municipality of Paese (CA2 and IR2); the third pair (3) was included in the Quinto di Treviso municipality (CA3 and IR3) [27].

The soils of pairs 1 and 2 were Cutanic Luvisols or Aric Regosols [26] with the following characteristics: stones 24–35%; sand 36–40%, silt 38–34%, clay 26%; pH = 7.6; total carbonate = 0–4%; organic carbon = 1.79–2.56%. The soils of pair 3 were Haplic Cambisols [26], with the following characteristics: sand 55–40%, silt 31–40%, clay 14–20%; pH = 7.6; total carbonate = 1–0%; organic carbon = 1.15–0.89%; as well as Fluvic Cambisols [26], with the following characteristics: stones 6–2%; sand 26–47%, silt 48–27%, clay 26%; pH = 7.6; total carbonate = 2–0%; organic carbon = 1.40–1.66%. The climate is sub-humid, with annual rainfall of about 950 mm. All soil areas are well drained.

The percentages of land under rotational and continuous maize cultivation in the six scenarios are shown in Table 3. In scenarios CA2 and CA3, continuous maize cropping clearly predominated (86.6% and 84.1%, respectively), while in scenario CA1 the proportion of reseeding was just above 60%.

Insecticide use (directed at adult beetles) equaled zero in all IR scenarios and in about 20% of CA scenario plots. Insecticide treatments at seeding (microgranules and coating) were applied to about 80% of the plots in the CA scenarios and to 60% of plots in the IR scenarios. Of these, about 40% were microgranule insecticides, which have the potential to significantly reduce root damage (mainly tefluthrin) [28].

The common agronomic practices used in all fields studied were similar to those described in the Section 2.2.1; the predominant tillage method was conventional tillage based on ploughing. At least three fields were selected in the center of each area for crop monitoring and pest evaluation.

#### 2.2.3. WCR Population Density and Damage Estimation

Within each crop rotation scenario (IR1, CA1, IR2, CA2, IR3, CA3), the following parameters were evaluated: (i) plant density and damage by other soil pests; (ii) number of WCR larvae in root systems (2016 only); (iii) root damage index (Node Injury Scale 0–3 [24]); (iv) catches of adult WCR plants with yellow sticky traps (Pherocon^®^ AM, Trece Inc., Adair, OK, USA), with data from 3 traps per field for up to 6 weeks; (v) crop damage in terms of percentage of “goosenecked” or “lodged” plants, the same as for the root damage index. Data collection linked to each parameter was conducted as outlined below.

Plant stand density and damage from other soil pests were assessed at the 5–8 leaf stage. Two plots of two rows measuring 20 m in length were selected for each of the 5–10 fields monitored in each area (IR1, IR2, IR3, CH1, CH2, CH4); all plants were counted and divided into healthy plants, plants showing signs of damage by wireworms, and plants showing signs of damage by other soil pests (e.g., black cutworm).

##### WCR Density

To estimate larval WCR densities, five plants for each of the 5–10 (only 1 in IR3) fields monitored in each area (IR1, IR2, IR3, CH1, CH2, CH4) were selected between the second stage and third crop growth stage according to the Davis model [29] at the time of maximum larval presence (late May–early June). Root systems and associated soil samples (intact with the roots) from the middle rows of each plot were randomly sampled with a shovel (approximately 15 × 15 × 15 cm). Samples were placed in funnels (18 cm in diameter) and kept under shelter with an open side allowing winds to enter; no lights were used to accelerate the drying process. Under each funnel, a water-filled vial (2.5 cm in diameter) was attached to collect larvae that moved downward as the soil dried. In order to prevent soil from falling into the vial, a 10 × 10 cm net (0.5 cm bore) was placed just under the sample. The funnels were inspected and larvae collected every two days to determine species and developmental stages. The labelled samples were washed and observed for root damage as described below.

The abundance of WCR adults was estimated using yellow sticky Pherocon^®^ AM (PhAM) or (pair 3) PALs Csalomon^®^ (Csalomon Group, Plant Protection Institute, Hungarian Academy of Sciences, Budapest, Hungary) (without pheromones) traps, set out in each of the 3–9 (2 in IR1 2016) monitored fields per area (IR1, IR2, IR3, CH1, CH2, CH4) per year. Three traps were deployed per field and observed according to official monitoring methods for a 6 week (42 day) period (weekly inspection of traps with removal of any attached beetles). The period commenced using predicted pupation times according to the Davis model and based on first captures in PAL pheromone traps (at least 3 per area). Yellow sticky traps attracting beetles by color were placed at least 30 m from the field edge and inside the field. The traps were attached to maize stalks with cable ties at the height of the plant ear, or alternatively on a 1 m high wooden stake in the case of fields with smaller plants. To ensure trap efficacy, the traps were all re-set at least once after the third week or whenever their condition (e.g., drying or wearing of the glue, or clogging with dirt or captured insects) might have affected captures.

##### Root Damage

Relative levels of root damage were estimated based on root rating or plant lodging. Damage estimates were made after silking.

For root rating, 10 root systems from maize plants from the middle rows of the 4–7 monitored fields per monitored area (IR1, IR2, IR3, CH1, CH2, CH4) were randomly sampled with a shovel. The excavated maize root systems were taken to the farm center for washing; after washing, root damage was assessed using the Node Injury Scale 0–3 [25]. To estimate plant lodging, plants were counted in two subplots of two rows measuring 20 m that were randomly selected in each of the 10–40 monitored fields per area (IR1, IR2, IR3, CH1, CH2, CH4), excluding margins and anomalous areas; damaged plants were recorded and divided into plants with “gooseneck” symptoms and lodged plants (presented as totals in this paper); the percentage of damaged plants ((n° of damaged plants/healthy plants + damaged plants) × 100) per subplot was calculated and used for statistical analysis.

### 2.3. Climatic Conditions

Details of climatic variables during the experimental period can be found at the ARPAV site [30] for Italy and through the Croatian Meteorological and Hydrological Service [31] for Croatia.

### 2.4. Statistical Analysis

#### 2.4.1. WCR Risk Factors Study in Italy and Croatia

To determine the significant predictors of damage risk, the percentages of damaged plants (goosenecked and lodged) at the sampled sites were first dichotomized into ≤5% and >5% to eliminate border effects and to include in the analyses only those fields with significant WRC damage. Logistic regression was performed to estimate damage risk based on predictive variables related to adult beetle populations, agronomic practices, soil properties, and treatments. The analysis was conducted using a univariate approach in which each predictive factor was included in the model in a stepwise fashion [32]. Estimated least square means were calculated. Pairwise post-hoc comparisons between levels of factors were Bonferroni adjusted. The relative risk (RR) of damage and 95% confidence intervals (95% CI) were calculated. Significant factors identified in the univariate approach were entered into a multivariate model to test their overall significance while excluding overlap between predictors. This analysis was performed using SAS 9.4 (Institute Inc., Cary, NC, USA).

#### 2.4.2. Crop Rotation Study in Italy

We used parametric and non-parametric analyses to compare WCR densities and damage in maize fields under rotation with those in conventionally managed fields. Normally distributed data (Shapiro–Wilks test) and data that passed the Levene homoscedasticity test were analyzed using univariate ANOVA with two factors (scenario (levels = IR and CH) × year (levels = 2016 and 2017)) and post-hoc comparisons were conducted using the HSD–Tukey test. Parameters that did not meet the test conditions for ANOVA were analyzed using the non-parametric Kruskal–Wallis test (% damaged plants, nr. larvae/plant) and multiple pairwise comparisons were made using Dunn’s procedure. Pearson’s correlation was used to assess relations between the percentage of rotation and adults above threshold densities. All analyses were performed using IBM SPSS Statistics version 22.

## 3. Results

### 3.1. WCR Risk Factors Study in Italy and Croatia

Results for fields with visible plant damage (gooseneck + lodged plants) greater than 5% are presented in Table 4. In Italy, during the first 3 years of the survey (2010–2012), visible WCR damage was detected exclusively in continuous maize fields (with at least 6 years of previous continuous maize cultivation); this encompassed 37 ha comprising 18 damaged fields. In 2010, outbreaks were limited to the western part of the region (Verona and Vicenza Provinces); in the two following years, visibly damaged fields were also found in the central part of the region. In 2013, some damaged fields without a long history of continuous maize cultivation (some were in their second or third year under maize) were reported for the first time.

In 2014, a few fields with first-year maize damage were found (Table 5); they occurred exclusively in plots near maize fields with continuous cropping, where continuous maize fields comprised more than 50% of the cultivated area and where high to very high populations of adult WCR occurred (>6 adult/trap/week, mostly >10 adult/trap/week, often more than 20). Damage from WCR was also observed in Croatia in fields under the first year of maize, but significant root damage and lodging were limited to the margins, at up to 20 m from the boundary of the continuous maize fields [23].

Low WCR populations (<2 adult/day) had a low probability of causing maize damage; medium, high, and very high beetle populations (>6 adult/day) increased the risk of damage by 2.92-, 5.28-, and 7.97-fold, respectively, compared to low populations.

Most factors (including soil properties) did not affect the likelihood of maize damage, with the exception of sowing time. Late or very late sowing significantly reduced the risk of damage compared to normal sowing; in contrast, early sowing increased the damage risk by 1.21-fold.

Insecticide treatments against adults showed no effect in reducing the risk of WCR, while rotation C reduced the risk of damage compared to both rotation regimes A and B.

In-furrow insecticide treatments, all considered, did not reduce severe (gooseneck-lodging) plant damage risk by WCR larval feeding activity; however, whereas Ercole^®^ (lambda-cyhalothrin) treatment caused no risk reduction, in-furrow Force^®^ (tefluthrin) significantly reduced the risk by about 40% (RR = 0.56, 95% CI 0.37–0.85, *p* = 0.006). In contrast, seed treatments, including all active ingredients, caused a slight risk increase. Force^®^ as a seed treatment, considered separately, also caused a slight increase in risk.

#### Multifactorial Model

Multivariate analysis of factors highlighted some changes in the estimation of risk ratios for the independent contributions by variables included in the final model: high to very high WCR beetle populations continued to be a significant factor (*p* < 0.001), whereas medium beetle populations had a relatively lower impact (*p* = 0.03). Force^®^ (tefluthrin) used as an in-furrow treatment reduced the risk by about 50% (RR = 0.46; 95% CI 0.22–0.93; *p* = 0.03) with respect to no treatment, and late sowing tended to reduce the risk of severe damage by about 40% (RR = 0.56; 95% CI 0.31–0.99; *p* = 0.05) with respect to the ordinary sowing date. No other factors were statistically significant predictors of risk.

### 3.2. Crop Rotation Study in Italy

#### 3.2.1. Structural Crop Rotation

The first catches with PAL sex pheromone traps were recorded in 2010 (42 specimens, average of two traps placed in continuous maize fields), even though a first occurrence of WCR was expected in 2008 [17]. In 2011, the first catches with PhAM traps (0.5 beetles/trap/day over a 42 day period) were recorded in continuous maize fields. WCR populations increased in continuous maize fields until 2015, when PhAM traps reached 5.8 beetles/trap per day over a 42 day period in some fields. Over the next few years, general reductions in maize cultivation, as well as for continuous maize, probably caused large decreases in populations, which were consistently maintained at negligible levels (SR1, SR2, Figure 2); however, in areas with high proportions of continuous maize, WCR populations remained very high (CM1, CM2, Figure 2), even when PhAM traps had been set in first-year maize fields. Nevertheless, no WCR root or plant damage was observed. The same population patterns were observed in the other two experimental farms (Sasse Rami and Diana), with average beetle population levels never exceeding two individuals/trap/day. Visible WCR damage symptoms were never found at these two farms. Although having a large cultivated area, these two farms were not isolated like Vallevecchia; nevertheless, the WCR populations that colonized these farms in the end of the 2000s [17] showed no significant increase in density over the next 10 years.

#### 3.2.2. Flexible Crop Rotation

The plant densities were generally good in both survey years; between 85% and 90% of the seeds sown resulted in plants that developed regularly, while infestations by wireworms or other soil pests (such as black cutworms) were negligible in both soil-insecticide-treated and non-treated plots.

The Node Injury Scale 0–3 showed significant differences in the degree of root damage between scenarios, with higher values in CA than in IR, and a statistically significant difference between the two treatments in area 3 (Figure 3A), (F = 31.39; *p* < 0.001; N = 740), where the maize percentage of cultivated land was the highest.

The percentages of plants with gooseneck symptoms or that were lodged (i.e., WCR damage) were found to be significantly higher in the CA scenarios compared to the IR scenarios (Kruskal–Wallis *K* = 30.67; *p* < 0.001; N = 260) (Figure 3A). No damage to maize was observed in year 1.

Cumulative adult catches/trap/day in week 6 were significantly higher in the CA scenario than in the IR scenario in areas 2 and 3 (F = 12.17; *p* < 0.001; N = 201), where the prevalence of continuous maize was higher (>80% in CA); there was no significant difference in area 1, where there was less difference between the 1CA and 1IR rotational plots (Figure 3B).

Observations of larval abundance were made only in 2016 and showed significantly higher numbers of larvae in CA scenarios (1.254 larvae/plant) compared to IR scenarios (0.796 larvae/plant) (*K* = 4.503; *p* = 0.034; N = 199) (Figure 3C).

The percent of maize area that exceeded the damage threshold and that had an appreciable risk of yield reduction (≥6 beetles/trap/day) and the percent of rotated area were inversely correlated (Figure 4). This confirms the evidence from Szalai et al. [33], who found that the higher percentages of damage occurred in plots with less than 60% rotation; in our case, higher damage rates were found in the 1CA, 2CA, and 3CA scenarios (Figure 4), which had values of 38.1%, 13.4%, and 13.5% of the rotation area, respectively.

The risk of root damage to contiguous corn fields was nearly twice as high in the CA scenario (47%) as in the IR scenario, with the risk of overall crop damage being five times (80%) higher. This illustrates how adult pressure (density) in a given area determines the actual risk of maize damage under the same soil, agronomic, and climatic conditions, as pressure is positively associated with contiguous maize field density, which favors WCR reproduction.

## 4. Discussion

### 4.1. WCR Risk Factors Study in Italy and Croatia

As shown in Table 4, beetle population densities accounted for most of the risk of maize damage from WCR. Most methods used to assess WCR damage risks use the beetle population density as the main predictor [7,8,34]. Therefore, management actions that reduce WCR population growth rates are essential to reduce damage risks. Very high beetle populations may cause significant damage to maize, even in rotated fields where maize has not had a major presence. In addition to second-year maize fields, first-year maize fields may also be damaged (Table 5), although this is a rare occurrence. It is clear that high WCR populations in cultivated maize promotes the spread of beetles to neighboring maize fields, and occasionally to non-maize fields.

Insecticides (all active ingredients considered) applied as in-furrow applications or as seed coatings did not reduce the risks of severe damage to maize from WCR. Only in-furrow applications of tefluthrin resulted in an estimated 50% risk reduction. In previous studies, this insecticide also showed potential to reduce WCR damage to roots [28,35,36,37,38]. Contrary to expectations, univariate analyses indicated that insecticidal seed treatments slightly increased the damage risk. In any case, this factor was weak (not significant after multivariate analysis). The fact that high to very high populations of WCR were found in many more fields with insecticide-treated seeds than in untreated fields, together with the phytotoxicity of some active ingredients, which may have reduced plant resistance to lodging, might explain these negative impacts of seed treatments. A low efficacy of insecticide seed coatings has also been previously reported by several authors [35,37]. All other factors, including soil texture, maize hybrid, and insecticidal treatments against adults, played no significant role in reducing risks.

### 4.2. Crop Rotation Study in Italy

Despite the high use of soil and foliar insecticides, larval and beetle densities were significantly higher in the CA (with prevalence of continuous maize and higher insecticide use) scenario than in the IR (high rotation rate and lower insecticide application) scenario. The build-up of WCR beetle populations to densities at which they pose a high risk of damage to maize crops took several years. After a successful eradication program (Venice airport area) that started in 1999 [17], WCR populations in Veneto were negligible until 2005. Afterwards, there was a progressive establishment of populations from the west (Lombardy) and east (Friuli Venezia Giulia), as shown by the use of pheromone traps [39], while no WCR beetles were found in the first eradication focus area in Venice. The spread of WCR from the east and west increased its presence in the region. Population densities continued to increase over several years, with the first visible WCR damage to maize occurring in 2010 in the western part of the region (Verona and Vicenza provinces) [40].

In Veneto, during the first 3 years (2010–2012), visible WCR damage was found exclusively in continuous maize fields (with at least 6 years of previous continuous maize cultivation). In 2013, for the first time, some damaged fields were found where there was no continuous maize cultivation (e.g., some maize fields in the second or third year). Beginning in 2014, when most fields were found to have high beetle densities, many damaged maize fields were observed that did not have a long history of continuous corn planting, including some first-year maize fields (Table 5). This limited number of fields with first-year maize damage (Table 5) were located exclusively in plots near maize fields with continuous cropping, in areas where continuous maize fields comprised more than 50% of the cultivated area, and where high or very high populations of adult WCR occurred. These results are consistent with the “lack of maize fidelity” hypothesis [16,41] as a general mechanism underlying rotational resistance.

The results also support reports by authors from Croatia [23] that indicated damage during the first year of maize cultivation. These authors reported that maize can be damaged in the first year if the field is adjacent to a continuous maize field that had a high adult population in the previous year, and that damage is possible up to 15 m from the border. Kos [42] demonstrated that the type of the previous year’s crop had an effect on damage when adjacent to maize fields; fields seeded with sunflower had higher adult populations than fields seeded with soybean, sugar beet, and wheat. However, damage at increasing distances from the edge of the maize field in the first year after sowing the different crops did not differ. In Croatia, economic losses due to the feeding by WCR larvae on maize roots can be expected in first-year fields (up to 10 m from the edge of the “donor” fields) only if WCR beetle populations have reached very high levels (weekly catches in the donor fields at least 70 beetles/PhAM; i.e., 10 beetles/PhAM/day) [42].

The situation observed in Italy in 2014 differed from that in Croatia. The fields in Italy were generally 30–40 m wide and damage was widespread throughout the fields, including in the center and not only at the edge. Some fields were not bordered by maize fields in the year before damage. In contrast to Croatia, where the population was high in only a few continuous maize fields and there were not many continuous maize fields in the study area, in Italy continuous maize fields were prevalent, supplying a large radius of available food and promoting large beetle populations. In subsequent years, probably as a result of a sharp decline in maize fields in the area (from 30 to 50%), no further first-year damaged fields were detected. A dramatic decrease in maize acreage (by 31% on average) was observed throughout the Veneto region [43] (Figure 5), resulting in an overall decrease in the WCR beetle population. Awareness of the damage potential of WCR after visible damage was observed in some fields and increasing mycotoxin problems (high contents of aflatoxins during drought and in warm summers) led to a reduction in the price of maize grain and a consequent significant reduction in maize acreage.

The results from Italy may also be indicative of an established rotation-resistant strain that stopped causing severe damage in cornfields due to population declines below the damage threshold. Spencer and Levine 2006 [41] and Knolhoff et al. 2006 [44] described rotation-resistant beetles as more active and better fliers than wild-type beetles. Mikac et al. [45] showed that rotation-adapted beetles have wider wings, while Kadoić Balaško et al. [46] found that rotation-adapted WCR populations have a more stable and elongated wing shape, which means that these individuals can fly longer distances. Elongated wings are more aerodynamic and are considered part of the migratory movement, and could be a useful invasive dispersal strategy for mated females [45]. This behavior is simply an intensification of a solid genetically-based species trait that has been highly expressed wherever the species has invaded new areas from distances greater than 10 km per year. This has been observed in a population that is the result of a “unique” hybridization between WCR populations of different origins (including in the U.S., where rotation-resistant populations occur) and later European populations [12,13,14,15,18,47]. Although an increased ability of beetles to digest soybean leaf tissue has been noted in rotation-resistant populations [48], the increased propensity to escape and disperse to any field (lack of fidelity to maize), possibly triggered by the enormous competition at extremely high WCR beetle populations, seems to be the key factor explaining the first-year maize damage observed in north-eastern Italy. Fortunately, this phenomenon is reversible when WCR populations decrease in response to reductions in maize acreage.

In any case, structural and flexible crop rotation regimes both proved to be effective in keeping populations below the damage threshold. This study supports the conclusions of Szalai et al. [33] that higher damage occurs in plots with less than 60% rotation. Crop rotation has likely fragmented populations and forced them into bottlenecks, resulting in large reductions in population size [49,50,51].

This study mainly focused on maize fields damaged by WCR and the surrounding fields. Most of the other maize fields in the areas where damaged fields were located did not show visible WCR damage symptoms. At least 100 undamaged fields were found around each damaged field. This ratio confirms data from Agrifondo Mutualistico Veneto and Friuli, which introduced insurance coverage through the Mutual Fund (MF) [52,53]. Applications for compensation for WCR damage represented far less than the 0.5% of the total maize area covered by the innovative insurance instrument.

## 5. Conclusions

The survey of key agronomic–cultivation factors and beetle population levels in fields damaged by WCR allowed us to determine that beetle population densities accounted for most of the risk of maize damage from WCR. Area-wide studies showed that beetle population levels depend on the number of maize cultivation fields and rotation regimes applied; higher beetle populations occurred in plots with less than 60% rotation.

Insecticide use did not affect WCR populations, while cultural controls based on rotation significantly reduced the damage risk. Structural rotation consistently prevents WCR from damaging maize crops, while flexible rotation may introduce a slightly higher risk of local sporadic damage, as this means that the population is approaching the threshold for control actions.

Maize growers and decision makers must consider practical values (as reported here) for the percentages of maize and continuous maize in terms of acreage as key parameters on which to base effective WCR management.

Moreover, diversified crop rotation strategies that significantly impact pest populations may reduce the risk of pest adaptation. These strategies are also more convenient than costly genetic approaches (WCR GMO resistant hybrids) that can produce pest-resistant strains in a short period of time [4]. The rotational strategies we propose are likely to be valid wherever they are applied, with possible local adaptations.

The results of our study confirm that crop rotation is an effective strategy to keep WCR populations permanently below the damage threshold, thereby preventing the use of pesticides in compliance with current European legislation.

## Figures and Tables

**Figure 1 insects-13-00415-f001:**
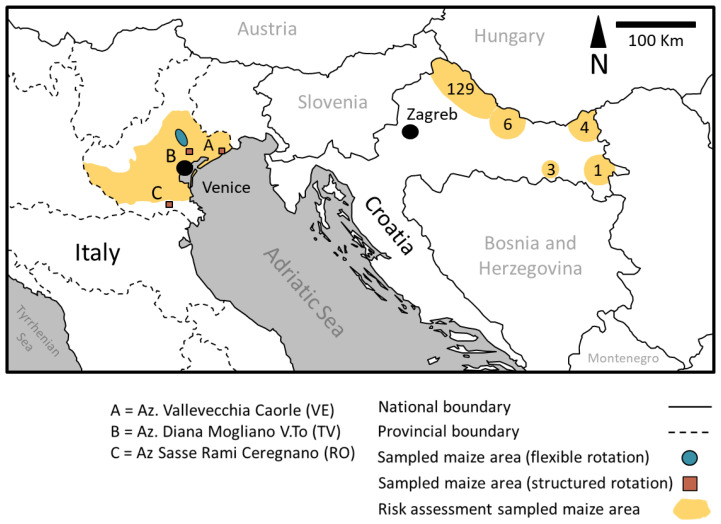
Map showing the regions in Italy and Croatia surveyed for WCR damage between 2003 and 2017. The locations of Venice and Zagreb are indicated by black circles. Orange coloring indicates areas surveyed for risk factors in Italy and Croatia. The numbers associated with surveyed areas in Croatia indicate the numbers of fields that were monitored. In Veneto, Italy, monitored fields were distributed throughout the entire maize cultivation area. Red symbols indicate the areas in Italy where farmers applied structural crop rotation and the blue symbol indicates the area where farmers applied flexible crop rotation. In Croatia, no comparisons of crop rotation regimes were conducted.

**Figure 2 insects-13-00415-f002:**
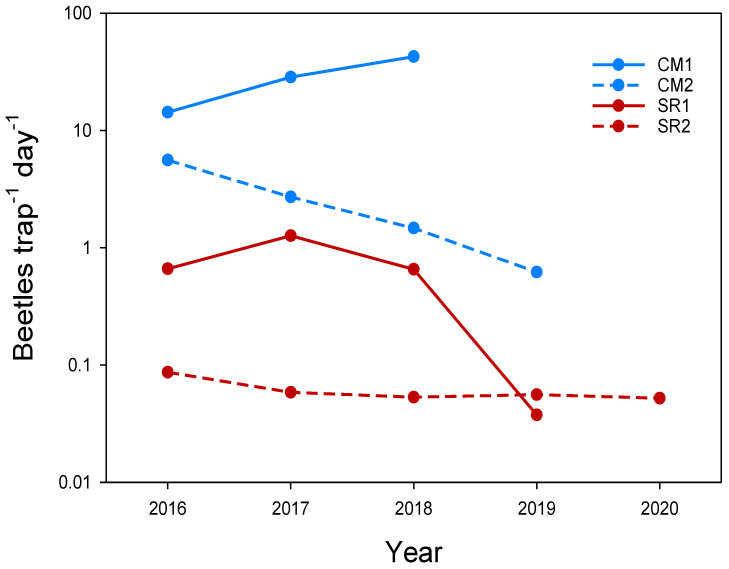
Effects of structural rotation (winter wheat/maize/soybean) on WCR population levels over the years. Numbers of WCR adults/trap/day (total sum at six weeks) in two different scenarios are shown—the first with a high percentage of continuous maize (blue lines and symbols = CM1, CM2) in Treviso province and the second based on structural rotation (red lines and symbols = SR1, SR2) at Vallevecchia pilot farm in Venice province (2016–2020), both in north-eastern Italy. Legend: CM1 = high presence of continuous maize with traps in continuous maize fields (Treviso); CM2 = high presence of continuous maize with traps in rotated maize fields (Treviso); SR1 = extensive structural crop rotation with traps in continuous maize fields (Vallevecchia, Venice); SR2 = extensive structural crop rotation with traps in first-year maize (Vallevecchia, Venice). Numbers are two-point moving averages.

**Figure 3 insects-13-00415-f003:**
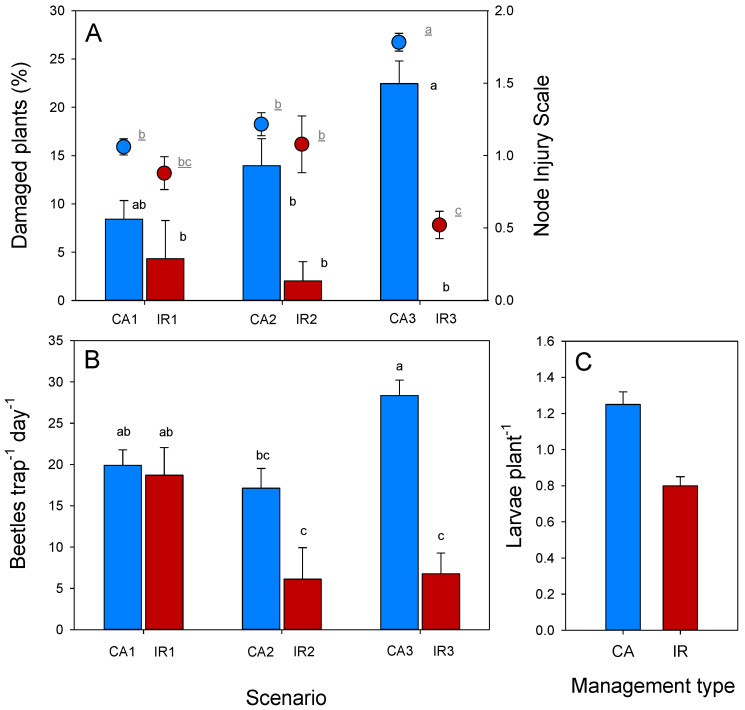
WCR pressure in the surveyed areas with continuous (blue) and rotated (red) maize crops: (**A**) Iowa index scores (Oleson: 0–3) and total (with gooseneck symptoms and lodged) damaged plants (%); (**B**) number of beetles captured by PhAM trap/day (average after six weeks), in the six scenarios under study (three areas hosting the two managements: CA, IR), years 2016–2017; (**C**) number of larvae per plant in the two managements areas: CA = chemical approach; IR = intensive rotation. Standard errors are indicated.

**Figure 4 insects-13-00415-f004:**
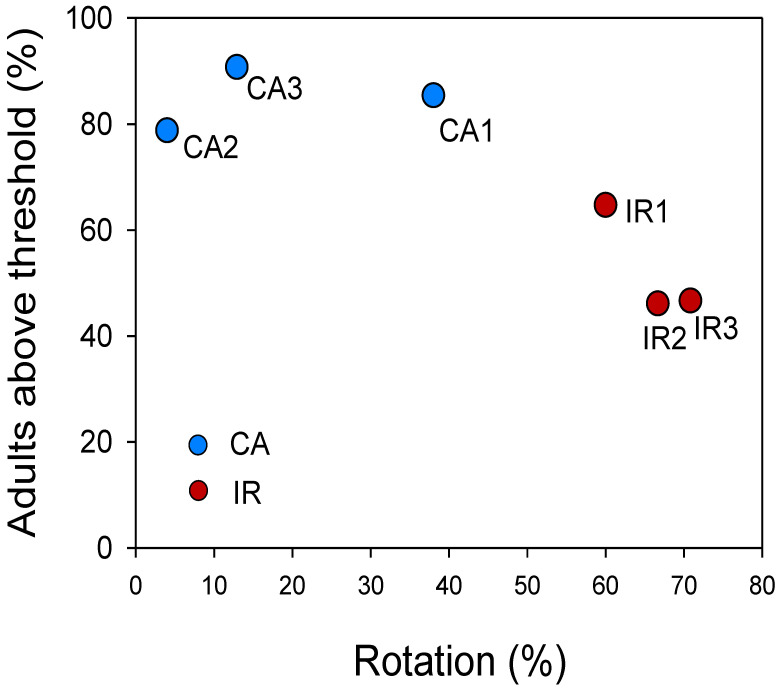
Relation between the percentage of cultivated land in rotation (no continuous maize) and the percentage of land where adult WCR levels exceeded the damage threshold (i.e., >6 beetles/trap/day) accumulated in week 6. CA = chemical approach; IR = intensive rotation. Numbers are scenarios indicated in Figure 3.

**Figure 5 insects-13-00415-f005:**
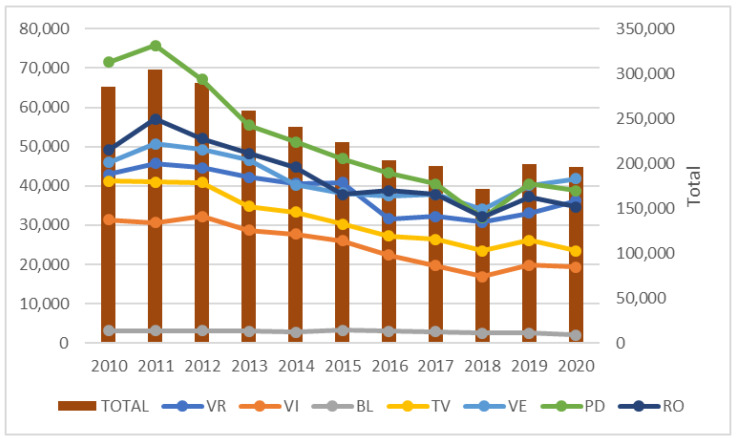
Maize cultivation land (ha) areas in the Veneto provinces from 2010 to 2020 (Source: Veneto Agricoltura 2021 [43]) (VR: Verona; VI: Vicenza; BL: Belluno; TV: Treviso; VE: Venezia; PD: Padova; RO: Rovigo).

**Table 1 insects-13-00415-t001:** List of the variables included in the database.

Variables	Explanation	Type	Classification ^1^	Number of Records		Surveyed Maize-Cultivated Land (ha)
				Croatia	Italy	
Year	Year of data collection	Ordinal	2003–2005	6	0	10
			2006–2010	60	5	62
			2011–2015	77	449	790
			2016–2018	0	148	393
Crop damage	Damage index: percentage of total plants damaged (gooseneck and lodged plants %)	Quantitative	<5%	98	345	864
			≥5%	45	257	391
WCR beetle population level PhAM Y-1	Beetle population level (from pheromone trapping) in the previous year	Qualitative	Low	23	65	137
			Medium	72	83	204
			High	28	110	241
			Very high	20	152	368
			NA *	0	192	305
Soil properties	Texture	Qualitative	F *	0	223	436
			FL **	134	21	216
			FA ***, FLA ****	1	207	269
			FS *****	0	49	117
			NA	8	102	217
Agronomic practices	Rotation type (see Section 2.1.3)	Qualitative	A (100% maize)	21	377	768
			B (70–80% maize)	116	183	416
			C (30–40% maize)	6	42	71
	Sowing date	Qualitative	Early (March)	1	226	414
			Ordinary (April–mid May)	139	240	619
			Late (second half of May)	3	100	143
			Very late (June)	0	36	79
	Hybrid	Qualitative	Dekalb	3	69	140
			KWS	0	37	54
			Pioneer	50	311	595
			BC Institute	84	0	81
			Others	6	67	125
			NA	0	118	260
Treatments	Insecticide treatments against adults in previous year	Qualitative	No	143	444	1022
			Yes	0	147	177
			NA	0	11	56
	Soil insecticide application	Qualitative	No	103	123	354
			In furrow micro-granular insecticide	0	306	510
			Insecticide coating	40	128	329
			Both (granular and coating)	5	27	43
			NA	0	13	19
	Insecticide (active ingredient)	Qualitative	No	0	39	354
			Tefluthrin in furrow	0	22	346
			Teflutrin as seed treatment	0	72	131
			Lambda-cyhalothrin	0	41	
			Others	0	35	

Note: ^1^: NA * = not available; F * = loam, clay loam; FL ** = loamy sand, sandy clay loam, silt loam, silty clay loam; FA *** = clay loam; FLA **** = silty clay loam, silty clay, FS ***** = sandy loam.

**Table 2 insects-13-00415-t002:** List of soil insecticides used on the fields during the years of monitoring.

Country (Number of Fields)	Product	Active Ingredients	Dose	Type
Croatia (14)	Cruiser^®^	Thiametoxam	0.63 mg/seed	coating
Croatia (3)	Gaucho^®^	Imidacloprid	1.2 mg/seed	coating
Croatia (3)	Macho^®^			
Croatia (18)	Poncho^®^	Chlothianidin	1 mg/seed	coating
Croatia (1)	Mesurol FS500^®^	Methiocarb 50%	1.8 L/100 kg of seed	coating
Croatia (6)	Force 1.5G^®^	Tefluthrin 1.5%	10–12 kg/ha	granules applied in-furrow
Italy (52)	Force ST^®^	Tefluthrin	0.5 mg/seed	coating granules applied in-furrow
Italy (91)	Force 0.5 G^®^	Tefluthrin	10–12 kg/ha	
Italy (17)	Poncho^®^	Clothianidin	0.5 mg/seed	Coating granules applied in-furrow coating
Italy (77)	Santana^®^	Clothiadinin 0.7%	11 kg/ha	
Italy (6)	Gaucho^®^	Imidacloprid	1.2 mg/seed	

**Table 3 insects-13-00415-t003:** Distribution of the rates of cultivated surfaces based on the number of consecutive years of maize cultivation in each rotation scenario under study, province of Treviso.

Scenario	Townland	Surface (ha)	No Maize	Maize							Number of Plots
				1st Year	2nd Year	3rd Year	4th Year	5th Year	6th Year	2nd–6th Year	
IR1	Paese/Trevignano	26.8	30.1	37.7	10.6	6.5	1.5	1.9	11.6	32.1	17
CA1	Montebelluna/Trevignano	30.6	19.8	18.3	1.6	7.2	17.9	1.8	33.2	61.9	54
IR2	Paese	27.3	57.9	16.0	6.9	1.9	0.0	3.5	13.7	26.0	13
CA2	Paese	17.0	0.0	13.4	0.0	6.3	6.8	5.6	67.9	86.6	33
IR3	Quinto TV	15.2	37.9	34.6	23.3	0.0	4.2	0.0	0.0	27.5	30
CA3	Treviso/Quinto/Paese	23.2	9.0	4.5	14.7	8.6	0.0	0.0	60.8	84.1	54

**Table 4 insects-13-00415-t004:** Univariate risk analysis establishing the risk of plant damage (gooseneck + lodged plants) exceeding 5%.

Variables	Level	LS Means: % Cases for Total Damage ≥5%	Comparisons	*p*	RR 95%CI
WCR beetle population level PhAM Y-1	Low	8	reference level		
	Medium	23	M vs. L	0.006	2.92 (1.36–6.28)
	High	42	H vs. L	<0.001	5.28 (2.53–11.04)
	Very high	63	VH vs. L	<0.001	7.97 (3.88–16.36)
Soil texture	FS *	26	reference level		
	F **	38	F vs. FS	0.994	
	FA ***	37	FA vs. FS	0.990	
	FL **** + FLA *****	32	FL + FLA vs. FS	0.990	
Rotation type	A (100% maize)	23	C vs. A	0.038	0.57 (0.33–0.97)
	B (70–80% maize)	40	C vs. B	0.017	0.52 (0.31–0.89)
	C (30–40% maize)	44	reference level		
Sowing date	Early (March)	41	early vs. ord.	0.039	1.21 (1.01–1.44)
	Ordinary (April–mid May)	50	reference level		
	Late (second half of May) + Very late (June)	25	very late + late vs. Ordinary	0.002	0.61 (0.45–0.84)
Hybrid variety (producer)	Dekalb	22	De Kalb vs. KWS	0.142	
	KWS	36	reference level		
	Pioneer	40	Pioneer vs. KWS	0.053	
	BC Institute	32	BC Institute vs. KWS	0.259	
	Others	55	Other vs. KWS	0.005	2.53 (1.32–4.84)
Insecticide treatment against adults in previous year	No	40	reference level		
	Yes	43	yes vs. no	0.527	
Soil insecticide application	No	39	reference level		
	In furrow micro-granular insecticide	57	seed vs. No	<0.001	1.45 (1.18–1.79)
	Insecticide coating	33	yes vs. No	0.109	
	Both (granular and coating)	31	yes + seed vs. no	0.400	
Insecticide (active ingredient)	No	39	reference level		
	Tefluthrin in furrow	22	teflutrin in furrow vs. No	0.006	0.56 (0.37–0.85)
	Teflutrin as seed treatment	72	tefluthrin seed treat. vs. No	<0.001	1.82 (1.44–2.30)
	Lambda-cyhalothrin	41	lambda-cyhalothrin vs. No	0.739	
	Other	35	other vs. No	0.325	

FS * = sandy loam, loamy sand, sandy clay loam; F ** = clay loam; FA *** = clay loam; FL **** = silt loam; FLA ***** = silty clay loam, silty clay.

**Table 5 insects-13-00415-t005:** Maize fields following no maize crop in the previous year, damaged (>5% of goosenecked or lodged plants) by WCR larvae from 2014 to 2017 in Italy.

Rotation	Fields	Previous Crop	Ha	Adult Population Density (Number of Beetles/Trap Per Day) ^1^
C	8	Alfalfa (4) Barley (4)	10.04	Very high *
C	1	Alfalfa	0.14	High **
C	1	Sorghum	1.6	NA ***
B	16	Soybean (2), Winter Wheat (10) Barley (3) Pumpkin (1)	47.46	Very high
B	6	Soybean (4), Winter Wheat (2)	5.77	High
B	8	Soybean (4), Winter Wheat (2) Barley (2)	5.98	NA
Total	40		70.99	

Note: ^1^: * Very high beetle population: >10 beetles/trap per year; ** High beetle population: >6 beetles/trap per year; *** NA = not available.

## Data Availability

The data presented in this study are available on reasonable request from the corresponding author.

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
