# Peer review of "Risk Assessment and Area-Wide Crop Rotation to Keep Western Corn Rootworm below Damage Thresholds and Avoid Insecticide Use in European Maize Production"

_insects, 2022, doi:10.3390/insects13050415_

Round 1

Reviewer 1 Report

I liked the manuscript by Furlan et al and believe that it should be reconsidered as a new submission after addressing a number of major issues.  Overall, the writing is fine.  The primary issue is that the presentation of the data was so sloppily done that we do not know what the data are.  In addition, the experimental designs are not clear, so we do not know whether or not the statistics were done correctly.  These things need to be made clear prior to resubmission as a new manuscript.

Additional items

Line 55 – neither the title nor the text above this suggest the current manuscript is specific to Europe, yet nearly all information not gathered or written in Europe is ignored.  The cited manuscript specifically states that it summarizes only information from Europe and literally dozens of high-quality works from the U.S. are ignored to support the above statement.

Line 90 – fix the typing error of ‘Muller’ to be ‘Miller’

Lines 158 and 160 – eliminate the redundancy where “the main” is repeated.

Line 191 – lodging and goose necking can be common without any rootworm damage when there is strong wind and excess moisture.  Root damage ratings are the only way to be certain of rootworm damage, but this is not mentioned in this section.

Line 193 – subplots of what??? The main plot and experimental design have not been defined.

Line 200 – presumably these things were recorded for each subplot, but this is not clear and should be stated.

Line 202 – gooseneck plants are defined, but lodged are not

Line 260 – did the soil in the funnel have a heat source such as a light bulb to speed drying of the soil? If so, say so? What was the size and source of the funnel? What was the size and source of the vial at the bottom of the funnel? Were the vial and funnel connected?  Was there screen in the funnel to stop soil from falling into the vial? If so, what was the size and source of the screen?

Line 271 – what was the color of the traps?

Line 315-316 – what is meant by ‘root sampling’?  More information is required to know what is meant.  Were they washed and rated for damage? If so, say so and include the damage scale.

Line 367 – the experimental design is not clear for any of the experiments.  The statistical analysis section cannot be known to be correct unless the experimental design is defined and known.

Table 4 – this is a nice table, but it is meaningless without also presentations of the data that are clear.

Figure 2 – cannot read what is written due overlapping letters and numbers in the same space.  Fix.

Figure 3 – there are no numbers on the Y axis!  The graphs are too messy to determine what is going on.  It is the author’s responsibility to make this clear!  It most certainly is not clear at this point.

Author Response

Dear reviewer
Thank you for your valuable comments and suggestions. You have enabled us to improve the manuscript considerably.
We are pleased to inform you that we have been able to address all of your comments. The details are visible in attached word document.
For clarity, your original comments and suggestions are in black text, while our response is in blue text.
Please note that all changes in the manuscript are shown in "track changes" mode. However, there is also a "black version" attached. In this version all changes have already been accepted.
If we have inadvertently missed an issue, please let us know so we can correct it.
Kind regards
Authors

Reviewer 2 Report

Dear Authors, 

Thanks for that interesting manuscript and your persistent "breath" for that long term study proofing all this various parameters/variables influencing  WCR population, specifically provide a large dataset for proofing that crop rotation positively influence WCR decrease and risk. 

The manuscript contains a lot of data from various  parameters/variables/locations etc... that means that a really good overview plus structure of the chapters and tables are needed for somebody, reading the first time the manuscript and tries to follow each specific trial in detail. Therefore it also could make sense to divide that into two papers. One for Risk assessment  (Italy and Croatia) and the other one specifically on crop rotation scenarios (Italy, flexible/structured rotation). 

If you do all in one paper I  would recommend a strict structure, always using same naming (table / chapter in method and in results etc...), otherwise it is sometimes not clear. I added some comments into the attached word file. 

 Thanks a lot, 

Best wishes! 

Author Response

Dear reviewer
Thank you for your valuable comments and suggestions. You have enabled us to improve the manuscript considerably.
We are pleased to inform you that we have been able to address all of your comments as it is visible from word document..
For clarity, your original comments and suggestions are in black text, while our response is in blue text.
Please note that all changes in the manuscript are shown in "track changes" mode. However, there is also a "black version" attached. In this version all changes have already been accepted.
If we have inadvertently missed an issue, please let us know so we can correct it.
Kind regards
Authors

Reviewer 3 Report

Dear Author
There are many queries to be answered in the manuscript.

Introduction is not considerable as it need high level improvement. Please take care during the editing that material included is relevant to your work or not. According to the topic, crop rotation helps in the manage of selected insects, ultimately helping in lessing the insecticides use. 
I am unable to understand that, why authors are putting their efforts a lot in the IPM. Please give general concepts of the IPM as in paragraph 2. 

Materials and Methods is almost appropriate, some comments are attached here in annotated file.

Results needs some attention according to the comments in the attached file.

Discussion is not valid for the article. Please do not repeat result. Please make it constructive according to the findings.
I do not think that this is valid discussion. Please read some relevant articles for the it.

Conclude your results here and future trend as well. Do not cite here. 

Author Response

Dear reviewer
Thank you for your valuable comments and suggestions. You have enabled us to improve the manuscript considerably.
We are pleased to inform you that we have been able to address all of your comments as it is visible from the attached document.
For clarity, your original comments and suggestions are in black text, while our response is in blue text.
Please note that all changes in the manuscript are shown in "track changes" mode. However, there is also a "black version" attached. In this version all changes have already been accepted.
If we have inadvertently missed an issue, please let us know so we can correct it.
Kind regards
Authors

Round 2

Reviewer 1 Report

As noted in my first review, I liked the manuscript.  Unfortunately, it does not appear that the authors even know what an experimental design is.  A randomized complete block is likely the most common in entomological studies.  Others include a completely randomized design, split-plot, split-split plot, repeated measures, factorial, etc.  In order to do statistical analyses correctly, the experimental design must be understood and explained to the reader.  It is now clear to me that there was no experimental design.  A series of data were collected and correlated, but we know little of the conditions in which they were collected.  Lots of information is presented.  The revised draft was thrown together so fast that comments from coauthors were included without addressing the concerns they raised.  As is, I am not sure what figures have been deleted versus added.  A clean version should be provided to make this clear.  Figure 5 is unreadable.  The references cited section has a mixture of every word being capitalized and those with caps only for the first word of a sentence. All should be converted to the destination journal format.

Additional items

Line 66 – ‘the’ before ‘most damage’.

Lines 270, 442, 483, 615, 649, Fig. 3a y axis - there is no “Iowa State University Scale” that goes from 0 to 3.  Long ago, where was an Iowa 1 to 6 scale, but the 0 to 3 scale is called the Node Injury Scale (see Oleson et al. 2005)

Lines 277, 296, 443, 466, 471 – not everything needs to be repeated each time.  What is Chromotropic?  In the U.S., we just call them yellow sticky traps.

Author Response

Dear Colleague, thank you very much for your appreciation.

In order to easy follow our answers we inserted your comments and the answers we prepared.

Your comment:

  • As noted in my first review, I liked the manuscript.  Unfortunately, it does not appear that the authors even know what an experimental design is.  A randomized complete block is likely the most common in entomological studies.  Others include a completely randomized design, split-plot, split-split plot, repeated measures, factorial, etc.  In order to do statistical analyses correctly, the experimental design must be understood and explained to the reader.  It is now clear to me that there was no experimental design.  A series of data were collected and correlated, but we know little of the conditions in which they were collected.  Lots of information is presented.

Our answer:

As to the criticism on the statistical approach, we have been working with an area-wide phenomenon that could not be studied with an “easy-to-manage” randomized block or split-split  layout on some thousands of square meters.  We had to work over hundreds of hectares. Behind the manuscript there was a huge effort, not less than 20,000 working hours in > 10 years; as to area wide experimentation in Veneto in 2016-2017  the effort was not less than 10,000 working hours in 3 years involving 10 people and dozens of farmers;  in 2015 a staff of 4 persons (see acknowledgements) including two officers of an important farmers’ organization was established; based on official data available (for EU subsidies) and interviews with farmers three (3) main areas (see table 3 area 1 Trevignano, area 2 Paese, area 3 Quinto di Treviso) were found that could be considered quite homogeneous based on the main agronomic characteristics  and where severe WCR damage had already occurred; based on database on rotation in each area we identified sub-areas having a “rotation approach” (IR, Intensive Rotation) and sub-areas having a prevalent “continuous maize approach (CA, Chemical Approach); in  this case we headed to areas with high livestock density. After this first step, we selected and interviewed many farmers to identify the areas where most farmers were following agricultural practices based on our main “ideal treatments” under study: i) intense rotation/low insecticide inputs (IR) and ii) continuous maize/high insecticide application (CA); see figures below. We may consider each main area (area 1, area 2, area 3), each including IR and CA approaches, a block with two treatments (and this was a first work hypothesis when we started the project); but we estimated that this classical approach, likewise a t-test, was not suitable; the possibilities to reduce variability working at a such large territorial level with many different actors playing a role inside, was much lower than the possibility you have while working with a classical plot layout at small scale level (being old I can assure you that I have been planning at least 500 classical plot layouts - mainly randomized blocks -  in my carrier, maybe I know what you mean); therefore we considered as first factor the scenario (combination between area 1,2,3 and IR or CA approach) and second factor the year (2016, 2017);

In effect our blocks are treated as individual experiments, each experiment replicated three times. We now clearly indicate that this is the case and have added (see highlighted text) the numbers of samples collected at these sites on each sampling occasion. The sampled fields were interspersed on each occasion, thereby avoiding issues around segregation of sampling. These explanation have now been added to the text.

Your comment:

The revised draft was thrown together so fast that comments from coauthors were included without addressing the concerns they raised. 

R: Due to very many comments and corrections, we overlooked some of the comments before submitting. We have now deleted all comments to make the version clearer.  

Your comment:

As is, I am not sure what figures have been deleted versus added.  A clean version should be provided to make this clear.  Figure 5 is unreadable. 

R: We are very sorry, but there may have been technical (software?) problems with the figures. The latest version should be viewed with markups turned off to see the revised positions of the graphs and the final versions of each graph.

Your comment:

The references cited section has a mixture of every word being capitalized and those with caps only for the first word of a sentence. All should be converted to the destination journal format.

R: We have now converted references to the journal format.

Specific comments:

Line 66 – ‘the’ before ‘most damage’.

R: Corrected

Lines 270, 442, 483, 615, 649, Fig. 3a y axis - there is no “Iowa State University Scale” that goes from 0 to 3.  Long ago, where was an Iowa 1 to 6 scale, but the 0 to 3 scale is called the Node Injury Scale (see Oleson et al. 2005)

R: You are right, although references are correct and avoid any misunderstanding; we are old enough to have been working many years using the 1-6 Hill and Peters root injury scale and many years with 0-3 Node Injury scale; we have corrected this now; thank you;

Lines 277, 296, 443, 466, 471 – not everything needs to be repeated each time.  What is Chromotropic?  In the U.S., we just call them yellow sticky traps.

R: chromotropic was intended to identify traps based on colour attraction to beetles, in contrast to traps based on pheromones, floral volatiles, etc.; we have changed chromotropic traps to yellow sticky traps and also tried to avoid all the repetitions.

Reviewer 2 Report

Dear Authors, 

Thanks for implementing the critical points in the previous proposal. The manuscript is now more clear and structured, therefore easier to follow (as it is such a comprehensive long-term study). 

Best wishes, 

Author Response

Dear reviewer

thank you for your comment. We are happy that you like the way how we improved the manuscript. However, based on comments of the reviewer 1 and 3 we did some additional improvements. We hope that you will be satisfied with the improvements too.

Kind regards

Authors

Reviewer 3 Report

Authors have put their potential in the revised version. But some changes or recommendations are still needed to be answer.

In the materials and Methods authors did not follow the comments, if they are not in favor of change, they must have to justify it in the comments response letter.

Figure 5 is still unacceptable.

Discussion improvement was not seen. In first 2, 3 paragraphs, authors should reduce the material and use cited material for the strength of results.

Author Response

Dear Colleague, thank you very much for your appreciation.

In order to easy follow our answers we inserted your comments and the answers we prepared. Our answers are in green.

Authors have put their potential in the revised version. But some changes or recommendations are still needed to be answer.

Thank you for the evaluation of the re-submitted manuscript. As we mentioned in our response we are happy to make further improvements and we are sorry if we did not answer on all comments, this was unintentional.

In the materials and Methods authors did not follow the comments, if they are not in favor of change, they must have to justify it in the comments response letter.

In the first round of review there were the following comments in section Methodology by reviewer 3 (line numbers corresponds with the line number in the version uploaded into the system at first submission and commented by reviewer 3 in the first round):

L 120-121. It is asked to delete the part of the sentence.

We deleted.

L147-149: The sentence “Sandy loam and loamy sand were classified as "light soils", loam and sandy clay loam as "loam soils", and silty clay loam, silt loam, clay, and clay loam as "heavy soils." shall be re-written.

R: you are right; now done; The sentence now is: “The soils were loam, clay loam (F), loamy sand, sandy clay loam; silt loam, silty clay loam (FL), clay loam (FA), silty clay loam, silty clay (FLA), and sandy loam (FS). For statistical analyses FA and FLA were grouped together.”

L281: It was a question related to sub-title.

Since we re-organized the section 2 we hope that this is much clear now: structural crop rotation is a part of the WCR risk assessment study conducted in Italy. It was carried out from 2010 and 2020.

L 371: It was asked to cite the reference for the sentence: Logistic regression was performed to estimate damage risk based on predictive variables related to adult beetle populations, agronomic practices, soil properties, and treatments.

R: Logistic analysis is a general statistical approach as analysis of variance when you have as dependent variable a binary response. We inserted some of our previous work in which the same analysis has been applied.

L380: factorial analyses- you asked to add how many factors were analysed.

R: We have added additional information and changed the text in accordance with the reviewer’s comment as follows: Normally distributed data (Shapiro-Wilks test) and data that passed the Levene homoscedasticity test were analyzed using univariate ANOVA with two factors (scenario × year) and post-hoc comparisons were conducted using HSD-Tukey test.

Figure 5 is still unacceptable.

R: It is corrected now

Discussion improvement was not seen. In first 2, 3 paragraphs, authors should reduce the material and use cited material for the strength of results.

R: For the Discussion section, we have cut all the redundant parts and some comments. We have also introduced further improvements. Since, as usual in an original scientific contribution, the results section just reports research outputs without comments, in the discussion section we have tried to discuss the most relevant outputs and try to give the most probable explanations for relevant/outstanding results; if we cut more, some relevant explanations will be missing; we would like to avoid further modifications that would weaken manuscript insights.  

Kind regards

Authors

Round 3

Reviewer 3 Report

Accepted Changes.